# Food Puree for Seniors: The Effects of XanFlax as a New Thickener on Physicochemical and Antioxidant Properties

**DOI:** 10.3390/foods10051100

**Published:** 2021-05-15

**Authors:** Chang Geun Lee, Youn Young Shim, Martin J. T. Reaney, Hye-Ja Chang

**Affiliations:** 1Department of Food Science and Nutrition, College of Science and Technology, Dankook University, Cheonan-si 31116, Chungnam, Korea; dwp0007@gmail.com; 2Department of Plant Sciences, University of Saskatchewan, Saskatoon, SK S7N 5A8, Canada; younyoung.shim@usask.ca (Y.Y.S.); martin.reaney@usask.ca (M.J.T.R.); 3Prairie Tide Diversified Inc., Saskatoon, SK S7J 0R1, Canada; 4Department of Integrative Biotechnology, Biomedical Institute for Convergence at SKKU (BICS), Sungkyunkwan University, Suwon 16419, Korea

**Keywords:** senior-friendly foods, puree, XanFlax, viscosity thickener, antioxidant activity

## Abstract

With the increasing number of older adults, the elderly-friendly food market has been rapidly growing. The physicochemical and antioxidant properties of soymilk-based banana-blueberry-puree with and without flaxseed-based (XanFlax) and xanthan-gum-based (brand G) thickeners were compared as a potential senior food. Samples included a control, three treatments with XanFlax (1%, 3%, and 5%), and three treatments with brand G (1.35%, 2.7%, and 5.4%). The physicochemical (color, sugar, salinity, pH, viscosity, and hardness) and antioxidant properties [DPPH, ABTS, reducing power (RP), and total polyphenol content (TPC)] were compared. The chromaticity values (*L**, *a**, and *b**) and pHs were similar among all treatments and the control, but the salinity of brand G showed statistical differences (*p* < 0.05). All samples met the Korean Industrial Standards for senior foods in terms of viscosity and hardness, while samples with brand G were harder and more viscous than those with XanFlax and the control (*p* < 0.001). XanFlax samples had greater ABTS radical scavenging activities than the control and brand G samples (*p* < 0.001). Although, the developed puree can be a possible senior food product without the addition of thickeners, XanFlax might be applied as a non-xanthan gum-based viscosity thickener with antioxidant functions for senior-friendly foods.

## 1. Introduction

Population aging is a global phenomenon. Virtually every country globally is seeing an increasing number and proportion of older people as a portion of their population. The proportion of the population aged 65 and over has increased from 6% in 1990 to 9% in 2019 [1]. This proportion is expected to increase to 16% by 2050, with 1 in 6 people globally expected to be 65 years or older. In particular, the population aging rate is fastest in East Asia and Southeast Asia. Nine out of the 10 countries with the largest increase in the percentage of seniors worldwide between 2019 and 2050 will be in Southeast Asia [1]. The biggest increases will occur in the Republic of Korea (23.0%), Singapore (20.9%), and Taiwan (19.9%). Spain will be the only European country included with the ten countries with the largest increase in the elderly as a portion of the population by 2050 [2]. The proportion of the population aged 65 or older in Korea was 15.7% in 2020 [3]. According to the 2019 Korean National Health Statistics, the rate of oral function restriction among the elderly aged 65 and over was 29.2% for those aged 60 to 69 and was 44.0% for those aged 70 and over. The rate of complaints of discomfort during food mastication was 27.8% in individuals aged 60–69 and 41.0% in those 70 and over [4]. As aging progresses, teeth are lost due to dental caries and periodontal disease, resulting in mastication disorders, dysphagia, and digestive disorders. Neurological problems such as senility, stroke, Alzheimer’s, Parkinson’s disease, and throat cancer exacerbate dysphagia [5,6]. In addition, dysphagia might aggravate problems such as dystrophy, dehydration, and aspiration pneumonia, and it increases the risk of chest infection and even death [7]. Therefore, special attention should be paid to dysphagia patient diets, and health should be maintained with adequate nutrition to reduce nutritional disorders caused by eating discomfort [8,9].

For diets that mitigate dysphagia effects, it is typical to choose foods and ingredients that slow the passage of food through the esophagus. Viscosity-enhancing food additives (viscosity enhancers) for patients with dysphagia are recommended. In this period of aging, health management of the elderly has emerged as a social problem, and the need for developing elderly-friendly food as food for general elderly people rather than patients is urgently required. In recent years, the food industry has responded to the market growth potential for elderly-friendly foods and is spurring the further development of convenience foods for those that experience difficult swallowing [10].

Dysphagia diets comprise meals designed to prevent lung aspiration and to maintain proper nutrition by incrementally adjusting food viscosity according to the status and dietary adaptation of people with reduced swallowing abilities [11]. Texture-improved foods or viscosity modifiers are interventions used to slow the swallowing process [12]. Currently, in hospital meals in Korea, starch is added to dysphagia patient diets, or viscosity is adjusted using agar powder or another commercially available viscosity modifier. Pelletier studied the treatment of dysphagia, including a comparison of food preferences with the addition of five starch-based viscosity modifiers added at different concentrations [13]. More recently, the addition of gelatin to Gochujang stir-fried meat, and spinach sprouts were studied to determine the impact of chewing and swallowing [14]. In addition, gelatin was tested as an additive to chicken breast for diets of individuals with senility [15,16].

A puree is made by mixing dietary ingredients (vegetables, fruits, grains, meat, etc.), pulverizing the mixture, and straining it through a sieve to yield a liquid or paste, which is easy to masticate, swallow, and digest. If a food includes functional or healthy ingredients, it can serve to mitigate or prevent chronic disease. Such a puree can provide increased value to meal products designed for the elderly [17]. Banana (*Musa acuminate* Colla) is mostly eaten as raw fruit and is also used in cooking and processing. Ripe bananas contain 25% carbohydrates, are rich in vitamin A, vitamin C, and dietary fiber, and contain less fat than their calories. Bananas are also known to be effective in diets for individuals with various conditions and diseases such as diarrhea, appendicitis, uremia, nephritis, opus, high blood pressure, and heart disease. Among fruits imported to Korea, banana is the most consumed and is highly flexible as a food material [18]. Blueberry (*Vaccinium ashei*) is native to North America and contains many bioactive substances, including anthocyanins and carotenoid pigments. Research on the antioxidant, anti-diabetic, and anti-cancer activities includes that on preparing processed foods using blueberry ingredients [19,20,21,22]. A representative antioxidant-rich vegetable that contains sulfur-containing compounds such as ally propyl disulfide and flavonoid pigments such as quercetin and kaempferol is onion [23]. Onion (*Allium cepa* L.) compounds provide antioxidant, anti-cancer, and anti-bacterial effects and effectively prevent skin aging, inhibit lipid oxidation, and prevent lead poisoning [24,25]. The US National Dysphagia Diet (NDD) classifies foods according to three levels: from foods that do not require chewing, level 1, to other foods that need to be easy to chew and swallow, level 3. According to the NDD standard, puree foods that do not require chewing before swallowing belong to level 1 [26]. Therefore, we tested whether the banana-blueberry-puree met the standards of viscosity and hardness for the elderly foods with and without the commercially available viscosity enhancers.

A study of patients with dysphagia was conducted to compare commercial viscosity enhancers and grain flour as alternatives for the commercial viscosity enhancers [27]. Flaxseed (*Linum usitatissimum* L.) contains protein, minerals, omega-3 fatty acids, lignan, dietary fiber, vitamins, and nutrients necessary in modern diets, including tocopherol [28]. Flaxseed can be sold for food by removing toxic substances called cyanogenic glycosides during processing [28,29]. It is sold mostly in powdered form in Korea. In Canada and the United States, the pharmacological effects of flaxseed are considered proven, and various foods incorporate flaxseed and its products into health foods. For example, Prairie Tide Diversified Inc. (Saskatoon, SK, Canada) launched a high-dietary-fiber product with a flaxseed ingredient sold under the trademark name ‘XanFlax’. However, there is no research on the role of high dietary fiber products derived from flaxseed as viscosity enhancers for controlling food properties in Korea. Thus, XanFlax, in this study, was compared with a xanthan gum-based viscosity enhancer currently on the market.

The purpose of the study was to evaluate a soymilk-based banana-blueberry-purees that met or exceeded the standard of elderly-friendly foods in terms of hardness and viscosity. In addition, the physicochemical and antioxidant properties of commercially available viscosity thickeners used in the purees were compared to evaluate their utility as viscosity enhancers for the adjustment of senior-friendly properties.

## 2. Materials and Methods

### 2.1. Materials and Chemicals

All ingredients of puree were purchased through E-Mart (Cheonan, Chungcheongnam-do, Korea), and onion powder was produced in our laboratory. Commercially available viscosity enhancers, XanFlax (X, Prairie Tide Diversified Inc., Saskatoon, SK, Canada) and brand G as a xanthan gum-based viscosity modifier (G, dysphagia solution, Chungju, Chungcheongbuk-do, Korea), were used.

### 2.2. Puree Preparation

As presented in Table 1, the sample recipe was developed as an elderly-friendly food in our laboratory. After carrot and kale were blanched, the ingredients in Table 1 were blended (HR2067, Philips, Nogueira, Brazil) for 5 min to produce a puree sample. Subsequently, 100 g of puree samples was measured, and viscosity modifiers were added to each sample based on the manufacturer recommendations (Table 2). Samples with the viscosity modifier were then mixed for an additional 30 s. Each sample was held at room temperature of 22 °C for 3 min and was then remixed and used for experiments. A total of 7 samples, including the puree control, without the viscosity modifier, were used in the experiment: control, three samples adding 1 g (X1), 3 g (X3), and 5 g (X5) of XanFlax, and three samples adding 1.35 g (G1, syrup-like), 2.7 g (G2, yogurt-like), and 5.4 g (G3, pudding-like) of the commercial viscosity thickener according to the usage direction (stage 1, 2, and 3) based on 100 g of puree, respectively (Table 2).

### 2.3. Physicochemical Properties

Viscosity and hardness for quality standards for senior-friendly food were determined using the Korean Industrial Standards (KS) method, KS H 4897 [30].

#### 2.3.1. pH Measurement

Each sample (1 g) was diluted 10-fold, after which samples were homogenized for 60 s in a peristaltic homogenizer (AESAP1068, AES, Bruz, France). Samples were allowed to stand for 3 min, and the pH was recorded using a portable pH-meter (Testo 206, Testo Inc., Lenzkirch, Germany).

#### 2.3.2. Salinity Measurement

Salinity was measured on 5 g of sample. Each result presented is the average of three measurements with a salt/TDS-meter (HDS1024, Daeyoon Scale Industry Co., Ltd., Seoul, Korea).

#### 2.3.3. Color Measurement

The color of each sample (5 g) was measured three times using a colorimeter (RM200, LoviBond, Neu-Isenburg, Germany). The results were expressed using CIELAB parameters (*L**, *a**, and *b**); *L** (100 = white; 0 = black) is an indication of lightness; *a** measures chromaticity, with positive values indicating redness and negative values indicating greenness; and *b** measures chromaticity, with positive values indicating yellowness and negative values indicating blueness.

#### 2.3.4. Sugar Content

The sugar content was determined using a refractometer (Pal-1, Brix 0−53%, Atago Co., Tokyo, Japan) of a homogenized solution produced after diluting each sample 10-fold with distilled water followed by homogenizing (AESAP1068, AES, Bruz, France) for 60 s. The refractive index value obtained is reported as percent/degree Brix.

#### 2.3.5. Viscosity Measurement

The viscosity was tested with 500 mL of each sample and was determined using a digital rotary viscometer (RVDV-E, Brookfield Engineering Laboratories, Inc., Middleboro, MA, USA) at 20 °C and a shear rate of 12 rpm for 2 min. After measurements, the value converted by multiplying the corresponding coefficient by the corresponding value was expressed in mPa·s.

#### 2.3.6. Hardness

Hardness was measured using a physical property analyzer (Texture analyzer, CT3 10 K, Brookfield Engineering Laboratories, Middleboro, MA, USA). After filling samples to a height of 15 mm in a 40 mm-diameter container, measurements were performed at a compression rate of 10 mm/s, a clearance of 5 mm, and a sample temperature of 22 °C using a circular probe with a diameter of 20 mm. The value obtained by dividing the height (N) of the first peak by the area (m^2^) of the sample during the compression was interpreted as the hardness value (N/m^2^). This analysis was repeated five times, and three values, excluding the maximum and minimum values, were averaged and expressed.

### 2.4. Antioxidant Properties

#### 2.4.1. DPPH Radical Scavenging Activity

The 2,2-diphenyl-1-picrylhydrazyl (DPPH) radical scavenging activity was determined by adding 2 mg of DPPH to 50 mL of 99.9% ethanol to make a 100 μM DPPH solution, and the DPPH solution absorbance was measured using a spectrophotometer (SpectraMax iD3, Molecular Devices, San Jose, CA, USA) and was adjusted to 1.2 at 517 nm. Samples (0.2 mL) were added to 1 mL of 100 μM DPPH solution (Alfa Aesar, Incheon, Korea), they were covered with aluminum foil, and they were allowed to react for 30 min at 22 °C. Subsequently, the visible absorbance at 517 nm was measured using a UV spectrophotometer. The DPPH radical scavenging activity was converted by the following calculation (Equation (1)).
DPPH radical scavenging activity (%) = [(A_517_ control − A_517_ sample)/A_517_ control] × 100(1)
where A_517_ control is the absorbance of the control at 517 nm, and A_517_ sample is the absorbance of the sample at 517 nm.

#### 2.4.2. ABTS Radical Scavenging Activity

A stock solution of 12 mM of 2,2′-azino-bis (3-ethylbenzothiazoline-6-sulfonic acid (ABTS, Sigma-Aldrich, Gyeonggi-do, Korea) and 5 mM of potassium was prepared and then added to a potassium phosphate buffer solution (pH 7.4) at a ratio of 1:1, and it was stored in a dark room for 12 to 16 h. Thereafter, the absorbance was diluted with distilled water to obtain an absorbance of 0.70 at 734 nm before use. The sample (50 µL) and ABTS solution (1000 µL) were added and left for 15 min in a dark room, and the absorbance was measured at 734 nm using a spectrophotometer (SpectraMax iD3, Molecular Devices). The ABTS radical scavenging ability was converted by the following Equation (2).
ABTS radical scavenging activity (%) = [(A_734_ control − A_734_ sample)/A_734_ control] × 100(2)
where A_734_ control is the absorbance of the control at 734 nm, and A_734_ sample is the absorbance of the sample at 734 nm.

#### 2.4.3. Reducing Power

The reducing power (RP) was measured according to Oyaizu (1986) [31] with minor modifications. Briefly, each sample (100 µL) was used to prepare dilutions at various concentrations, and then each dilution was mixed with 500 µL of sodium phosphate buffer (200 mM, pH 6.6; Samchun, Seoul, Korea) and 500 µL of 1% potassium ferricyanide (Acros Organics, Seoul, Korea). The mixture was incubated in a 50 °C water bath (C-WBE-L, Chang Shin Science Co., Pocheon, Korea) for 20 min. Then, 500 µL of 10% trichloroacetic acid (Samchun) was added and centrifuged at 6000 rpm (1580 R, Labogene Co., Allerød, Denmark) for 5 min. The supernatant (500 µL) was mixed with 500 µL of distilled water and 100 µL of 0.1% ferric chloride, and then the absorbance at 700 nm was measured using a spectrophotometer (SpectraMax iD3, Molecular Devices). The RP was calculated by comparing it with the L-cysteine standard curve.

#### 2.4.4. Total Polyphenol Content (TPC)

Samples (0.1 mL) were mixed with 1.9 mL of distilled water, and 0.2 mL of Folin-Ciocalteu’s phenol reagent (Sigma-Aldrich, Gyeonggi-do, Korea) was then added. After 3 min at room temperature (22 °C), 0.4 mL of a saturated sodium carbonate solution (Na_2_CO_3_, Samchun Chemical, Seoul, Korea) and 1.9 mL of distilled water were added. The mixture was allowed to stand for 1 h in the dark at room temperature. The absorbance was measured at 725 nm using a spectrophotometer (SpectraMax iD3, Molecular Devices). Measurements were compared with a gallic acid standard curve and are expressed as mg/g of gallic acid equivalents in milligrams per gram (mg GAE/g).

### 2.5. Statistical Analysis

All statistical analyses were performed using the Statistical Package for the Social Sciences (SPSS) ver. 18.0 (IBM Corp., Somers, NY, USA). Mean comparisons were made by one-way analysis of variance (ANOVA) followed by the Bonferroni post hoc test. Data are presented as mean ± standard deviation (SD) (*n* = 3), and ** p* < 0.05 and *** p* < 0.001 were considered statistically significant.

## 3. Results and Discussion

### 3.1. pH, Salinity, and Sugar Content

Table 3 shows the pH, salinity, and sugar content of each sample added with commercially available viscosity enhancers XanFlax and brand G.

The pH values of the XanFlax and brand G purees were similar to that of the control (5.61 ± 0.09). The salinity of the control puree was 0.20%, those of the XanFlax purees were the same as the control, and those of the brand G purees ranged from 0.15% to 0.17%; the salinity decreased with the amount added (*p* < 0.05). The sugar content of the XanFlax purees (1.32 ± 0.04–1.35 ± 0.21%) was lower than that of the control (1.40 ± 0.30%), and the sugar content of the XanFlax purees decreased with the amount added; when 3% or more was added, there was no change in sugar content. The brand G purees showed higher sugar contents (1.58 ± 0.04–1.85 ± 0.05%) than the control did except for G1, and it was found that the sugar content increased with the amount added (*p* < 0.001).

### 3.2. Color Analysis

The chromaticity of each sample of soymilk-based banana-blueberry-puree showed that there were no significant differences in all of the values (*L**, *a**, and *b**) (Table 4). In the values of *a** and *b**, both the XanFlax-added puree (3.40–3.83) and the brand G-added puree (2.75–3.58) showed slightly higher values than the control did (2.63 ± 0.19). There was no pattern to the increase in *a** and the amount of XanFlax. The *b** value for the XanFlax X5 puree (16.25 ± 0.43) had a higher value than the control did (14.30 ± 1.89), but there were no other statistically significant differences.

In the current study of purees containing fruits and vegetables of various colors, the amount of addition did not affect the color. However, when yogurt was used as the matrix, the amount of flaxseed powder added influenced sample the chromaticity [32]. This report describes the effects of flaxseed powder on the chromaticity of mixtures with yogurt; the *L** value decreased with the amount of flaxseed powder, while there was no significant difference in the *a** value [32]. However, the value of *b** increased with the amount of flaxseed power added [32].

### 3.3. Viscosity and Hardness Analysis

Both the XanFlax purees (7545–31,775 mPa·s) and the brand G purees (17,984–45,109 mPa·s) had higher viscosities than the control did (4475.33 mPa·s), with the brand G purees having a greater viscosity than those of the XanFlax purees with similar amounts of added thickener (Table 5). In the XanFlax purees, 3% XanFlax imparted a similar viscosity to 1.35% brand G, and a puree with 5% XanFlax showed a lower viscosity than 5.4% brand G and a higher viscosity than 2.7% brand G (*p* < 0.001). The viscosity standard for senior-friendly foods was 1500 mPa·s or more in Korea based on the Industrial Standardization Act (Ministry of Agriculture, Food and Rural Affairs, 2020), and it was found that all the samples currently tested are suitable for the standard for senior-friendly foods. The viscosity of yogurt containing 1%, 3% and 5% flaxseed flour increased with the amount of flaxseed flour, and the results of this experiment are similar [32]. The dietary fiber and protein composites formed in the purees with both brand G and XanFlax form viscous gels and increase water retention, making the puree matrix more viscous [33,34]. XanFlax has both protein and dietary fiber [35]. The viscosity was thought to partly increase with the water retention capacity. The fiber derived from flaxseed has a thickening property similar to that of gums, such as xanthan gum currently used commercially [36]. In this experiment, the viscosity of XanFlax purees showed less viscous features compared to xanthan gum thickener under similar concentrations. Therefore, a concentration of XanFlax of at least 3% or more was recommended for a viscosity enhancer to have a viscosity and hardness similar to those of brand G.

The hardness values of the XanFlax purees (197.92–458.33 N/m^2^) were higher than that of the control (154.17 N/m^2^), and the purees containing 1% and 3% XanFlax showed similar hardness values to G1. The puree to which 5% XanFlax was added showed intermediate hardness between G2 and G3. The hardness of the brand G-added puree (210.42–697.92 N/m^2^) was higher than that of the control, and it was found that the greater the amount of G brand viscosity-enhancing agent added, the higher the hardness value (*p* < 0.001). Korean regulations (Industrial Standardization Act) have been established to classify senior-friendly food hardness values (N/m^2^). Stage 1 (tooth intake) was classified into 500,000.00–50,000.00 N/m^2^, stage 2 (intake by gums) was classified into 50,000.00–20,000.00 N/m^2^, and stage 3 (intake by tongue) was classified into 20,000.00 N/m^2^ or less [30]. The hardness of soymilk-based banana-blueberry-puree ranged from 154.17 to 697.92 N/m^2^, indicating that it is a third-stage product that can be ingested with the tongue and does not require chewing.

From the previous studies measuring the viscosity of the sweet-pumpkin-chicken-puree for seniors by adding each type of commercially available viscosity enhancer, the hardness of brand G was 110.91–178.36 N/m^2^, and the hardness of brand N was 138.31–158.09 N/m^2^. As a result, the hardness increased with the amount of viscosity-enhancing agent added [37,38]. However, as the hardness measurement conditions in both this study and previous studies are different, it is difficult to make a clear comparison, but the result that the hardness increases with the amount of addition agree with the results of this study. We also verified that the banana-blueberry-puree without a thickener was similar in viscosity to the 1% XanFlax thickener puree, and it is classified as stage 3 in level of hardness according to the Korean regulation standard for senior-friendly food.

In this study, viscosity and hardness were measured in accordance with the standards of senior-friendly foods, but more data are necessary for texture studies that are used to evaluate solid food for senior or dysphagia patients who suffer from chewing and swallowing discomfort. Therefore, it is considered that foods that are more suitable for the senior can be developed according to the characteristics of the senior when foods are manufactured in consideration of not only the viscosity and hardness of the KS standard but also other textures. As a follow-up study, when developing a senior-friendly food, not only viscosity and hardness but also other texture measures should be studied.

### 3.4. Antioxidant Analysis

The DPPH radical scavenging activity of the control puree was 23.02%, those of the XanFlax purees ranged from 19.78% to 34.70%, and those of the brand G puree ranged from 7.27 to 13.87% (Table 6). The DPPH radical scavenging abilities of the 1% and 3% XanFlax puree were similar to that of the control, but the scavenging abilities were higher than that of the control when 5% was added. The radical scavenging activities of XanFlax addition samples were higher than those of the brand G addition samples (*p* < 0.001). For ABTS, the radical scavenging activities of the XanFlax purees (67.05–67.84%) were higher than that of the control (61.62%), but there was no significant increase in the antioxidant property with the increase in the concentration of XanFlax. For the brand G samples, the ABTS radical scavenging ability decreased with the amount added (*p* < 0.001). There were no statistically significant differences in RP between the control and any treatment puree (1.71–1.90 mM). The control TPC was 1.19 mg GAE/g, the XanFlax purees ranged from 1.14 to 1.64 mg GAE/g, and the brand G purees ranged from 1.05 to 1.22 mg GAE/g. The TPC of the control was similar to the X1, G1, and G2 samples, was lower than X3 and X5, and was higher than G3 (*p* < 0.001).

The antioxidant activity results are partially consistent with previous studies. In the study that added flaxseed powder to yogurt, the total phenol content, DPPH radical scavenging activity, and ABTS radical scavenging activity increased [32]. It was reported that the DPPH and ABTS radical scavenging activities in the antioxidant activity of pound cakes added with 2%, 4%, 6%, and 8% flaxseed powder increased compared to that of the control puree, and the antioxidant activity was statistically significantly increased according to the amount added [39]. In addition, the DPPH radical scavenging ability in a sponge cake matrix prepared with 5%, 10%, 15%, and 20% flaxseed powder increased with the addition of flaxseed powder. The DPPH radical scavenging ability increased statistically with the amount added [40]. These results suggest that the TPC, DPPH radical scavenging activity, and ABTS radical scavenging activity increased with the number of phenolic compounds, such as lignan in XanFlax, and the high dietary fiber of flaxseed. However, in our study with soymilk-based banana-blueberry-puree, XanFlax treatments improve antioxidant activity compared to the control and the brand G samples in terms of ABTS radical scavenging activity. Thus, more research is needed due to the lack of research on food containing flaxseed fiber.

## 4. Conclusions

Increasingly, viscosity modifiers are being used to overcome dysphagia, particularly senior food swallowing disorders. This study was conducted to evaluate the possibility of soymilk-based banana-blueberry-puree as a senior-friendly food and the effects of available commercial thickeners on the physicochemical and antioxidant properties of soymilk-based banana-blueberry-puree. Experimental groups were classified according to the type and concentration of the thickeners (XanFlax and brand G). There were no significant differences among the sample analysis of pH, color (*L**, *a**, *b**), and RP. All samples including the control met the Korean Industrial Standards for senior foods in terms of viscosity and hardness. All samples met the KS for senior foods in terms of viscosity and hardness, while samples with brand G were harder and more viscous than those with XanFlax and the control (*p* < 0.001). The puree with 5% of XanFlax had a greater DPPH and ABTS radical scavenging through antioxidant activity and a higher TPC level than the control and brand G samples (*p* < 0.001). At least 5% XanFlax can be applied as a non-xanthan gum-based viscosity thickener with higher ABTS and DPPH radical scavenging activity for senior-friendly foods. Those results are considered as a study of viscosity analysis on dysphagia, and they may be used as database data on viscosity by food type.

## Figures and Tables

**Table 1 foods-10-01100-t001:** The recipe of soymilk-based banana-blueberry-puree.

Ingredients	Quantity (g)	%
Banana	200	22.47
Blueberry	102	11.45
Soymilk	300	33.69
Onion powder	6.4	0.72
Carrot	100	11.23
Kale	82	9.21
Milk	100	11.23
Total	890.4	100

**Table 2 foods-10-01100-t002:** Condition of thickeners added based on 100 g of the puree.

**Control** **(Puree)**	**Thickeners (g)**					
	**X1**	**X3**	**X5**	**G1**	**G2**	**G3**
No thickener	1.0	3.0	5.0	1.35	2.7	5.4

X: XanFlax; G: brand G (G1: syrup-like; G2: yogurt-like; G3: pudding-like).

**Table 3 foods-10-01100-t003:** Changes in pH, salinity, and sweetness of soymilk-based banana-blueberry-puree by adding thickeners XanFlax and brand G.

Characteristics	Control	X1	X3	X5	G1	G2	G3
pH	5.61 ± 0.09	5.63 ± 0.02	5.68 ± 0.04	5.69 ± 0.11	5.59 ± 0.14	5.67 ± 0.14	5.70 ± 0.17
Salinity (%) *	0.20 ± 0.00 ^a^	0.20 ± 0.00 ^a^	0.20 ± 0.00 ^a^	0.20 ± 0.00 ^a^	0.17 ± 0.05 ^ab^	0.15 ± 0.05 ^b^	0.15 ± 0.05 ^b^
Sugar (Brix, %) **	1.40 ± 0.30 ^b^	1.35 ± 0.21 ^c^	1.32 ± 0.17 ^c^	1.32 ± 0.04 ^c^	1.35 ± 0.16 ^c^	1.58 ± 0.04 ^b^	1.85 ± 0.05 ^a^

Control: puree without thickeners, X1: puree added with XanFlax 1%, X3: puree added with XanFlax 3%, X5: puree added with XanFlax 5%, G1: puree added with brand G 1.35%, G2: puree added with brand G 2.7%, G3: puree added with brand G 5.4%. Values in each row followed by different lower-case superscript letters are significantly different at * *p* < 0.05 and ** *p* < 0.001 (one-way ANOVA, Bonferroni test).

**Table 4 foods-10-01100-t004:** Changes in the colors of soymilk-based banana-blueberry-puree by adding thickeners XanFlax and brand G.

Color	Control	X1	X3	X5	G1	G2	G3
*L**	31.90 ± 5.88	31.15 ± 6.24	30.18 ± 5.83	29.85 ± 5.83	31.45 ± 4.99	31.53 ± 1.27	30.67 ± 0.53
*a**	2.63 ± 0.19	3.83 ± 1.54	3.40 ± 0.63	3.58 ± 0.85	3.58 ± 1.70	2.75 ± 1.23	3.10 ± 1.14
*b**	14.30 ± 1.89	15.90 ± 3.06	15.42 ± 1.33	16.25 ± 0.43	15.43 ± 1.53	14.33 ± 1.53	15.25 ± 2.70

Control: puree without thickeners, X1: puree added with XanFlax 1%, X3: puree added with XanFlax 3%, X5: puree added with XanFlax 5%, G1: puree added with brand G 1.35%, G2: puree added with brand G 2.7%, G3: puree added with brand G 5.4%. *L**, brightness/darkness; *a**, (+) redness/(–) greenness; and *b**, (+) yellowness/(–) blueness.

**Table 5 foods-10-01100-t005:** Changes in viscosity and hardness of soymilk-based banana-blueberry-puree by adding thickeners XanFlax and brand G.

Characteristics	Control	X1	X3	X5	G1	G2	G3
Viscosity (mPa·s)	4475.33 ± 444.84 ^e^	7545.33 ± 1329.03 ^e^	16,162.33 ± 1433.48 ^d^	31,775.00 ± 5661.20 ^b^	17,984.33 ± 2785.55 ^d^	26,710.67 ± 2830.05 ^c^	45,109.67 ± 3547.27 ^a^
Hardness (N/m^2^)	154.17 ± 19.63 ^e^	197.92 ± 8.54 ^d^	212.50 ± 5.59 ^d^	458.33 ± 3.23 ^b^	210.42 ± 3.23 ^d^	347.92 ± 8.54 ^c^	697.92 ± 8.54 ^a^

Control: puree without thickeners, X1: puree added with XanFlax 1%, X3: puree added with XanFlax 3%, X5: puree added with XanFlax 5%, G1: puree added with brand G 1.35%, G2: puree added with brand G 2.7%, G3: puree added with brand G 5.4%. Values in each row followed by different lower-case superscript letters are significantly different at *p* < 0.001 (one-way ANOVA, Bonferroni test).

**Table 6 foods-10-01100-t006:** Changes in antioxidant properties of soymilk-based banana-blueberry-puree by adding thickeners XanFlax and brand G.

Characteristics	Control	X1	X3	X5	G1	G2	G3
DPPH (%)	23.02 ± 3.69 ^bc^	19.78 ± 4.21 ^c^	27.07 ± 2.00 ^b^	34.70 ± 1.20 ^a^	8.56 ± 1.80 ^e^	7.27 ± 1.67 ^e^	13.87 ± 3.06 ^d^
ABTS (%)	61.62 ± 0.4 ^b^	67.05 ± 0.38 ^a^	67.74 ± 0.58 ^a^	67.84 ± 0.49 ^a^	40.22 ± 1.56 ^c^	34.22 ± 0.93 ^d^	18.33 ± 1.10 ^e^
RP (L-Cysteine eq. mM)	1.76 ± 0.04	1.73 ± 0.02	1.76 ± 0.01	1.74 ± 0.08	1.76 ± 0.01	1.90 ± 0.43	1.71 ± 0.03
TPC (mg GAE/g)	1.19 ± 0.04 ^cd^	1.14 ± 0.03 ^d^	1.36 ± 0.03 ^b^	1.64 ± 0.04 ^a^	1.20 ± 0.02 ^cd^	1.22 ± 0.06 ^c^	1.05 ± 0.02 ^e^

Control: puree without thickeners, X1: puree added with XanFlax 1%, X3: puree added with XanFlax 3%, X5: puree added with XanFlax 5%, G1: puree added with brand G 1.35%, G2: puree added with brand G 2.7%, G3: puree added with brand G 5.4%. Values in each row followed by different lower-case superscript letters are significantly different at *p* < 0.001 (one-way ANOVA, Bonferroni test). RP: Reducing power; TPC: total polyphenol content; GAE: gallic acid equivalent.

## Data Availability

Data of the current study are available from the corresponding author on reasonable request.

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
