# Peer review of "Food Puree for Seniors: The Effects of XanFlax as a New Thickener on Physicochemical and Antioxidant Properties"

_foods, 2021, doi:10.3390/foods10051100_

Round 1
Reviewer 1 Report
Please rewrite abstract and conclusions stating significant effects and non-significant (similar) do no use terms terms high or lower etc.
L41-reword
All tables use superscrp a for the highest data b, c etc so on and be consistent.
L 265-266 Describe no effects first then compare with literature.
L 272 viscosity of x5 and G2.7 are not similar (T5).
L 284 conform to table data.
L 296-297 statement not correct. Discuss as stated for abstract and conclusions.
L 310-314 Comparing 20,000 or less N/m2 with 198-698 N/m2 ?
L 329-330 x3 > control not true.
L333 – reword correctly.
L 334 Incomplete sentence.
L 372-373 No consistent with T6.
Author Response
We appreciate all your comment and your insights. With the reflection of your review opinions, the revised parts were indicated with blue letters. Abstract and conclusions were also rewritten as your comments (Line 11-23, Line 374-381).
1) L41-reword => we revised as Line 36-42
2) All tables use superscript a for the highest data b, c etc so on and be consistent.
=> Table 3, 5 and 6 were revised as your comments.
3) L 265-266 Describe no effects first then compare with literature.
=> Revised as Line 261-264.
4) L 272 viscosity of x5 and G2.7 are not similar (T5).
=> Revised as Line 274-275.
5) L 284 conform to table data.
=> Revised as Line 286-289.
6) L 296-297 statement not correct. Discuss as stated for abstract and conclusions.
=> Revised as Line 300-301.
7) L 310-314 Comparing 20,000 or less N/m2 with 198-698 N/m2 ?
Please refer to Line 315-319.
8) L 329-330 x3 > control not true.
=> Revised as Line 332-333.
9) L333 – reword correctly
=> revised as Line 337-338.
10) L 334 Incomplete sentence.
=> Revised as Line 342-343.
11) L 372-373 No consistent with T6.
=> Revised as Ling 374-380.
Reviewer 2 Report
This paper focuses on the comparison between thickeners that can be used to formulate puree recipes that can positively handle dysphagia problems. The selected methods are appropriate, however, the work seems a bit confusing, both in the introduction and how the discussion of the results was performed. Written English should be revised in the sense that some sentences are awkwardly framed) Some issues that I think must be addressed are detailed below.
The Introduction seems all over the place. Is important to frame the subject at hand, however, the hypothesis tested should be better explained.
L91-92. Explain what is meant by saying "first stage dietary modification"?
How was the "main" recipe obtained/developed?
There is no information on the temperature conditions during the steps of processing the puree.
L120. "was" must be corrected to "were". Please verify the verb tense throughout the manuscript.
L119. Explain better the selection of different mass for the xanthan gum-based samples.
I think all the steps (in Figure 1) are described before in the text. If so, Figure 1, as it is, should be removed since is not adding new information. In this particular case, I would recommend adding a Scheme (more explicit) explaining in detail all the steps, since information like processing conditions are missing (i.e. mixing velocity; temperature etc..).
L161. The 500mL is referring to the tested volume for each sample.
L170. I am not sure if the designation "height" is correct. This refers to the maximum point of hardness (N) obtained, right?
L174. Is there any indication of the time after package for the commercial samples? Can this be a determinant factor regarding the comparison tests?
I am curious about the fluidity of the samples. Was the procedure mainstream as a result of the increased viscosity of some samples (like measuring 0.2 mL of sample)?
L268. Correct the (•) symbol in the entire manuscript.
L301-302. The decimal point form should be uniformized.
L303-304. The sentence is confusing and should be revised.
L325. The discussion of the results could be improved. G puree results are a result of poor levels of total phenolics?
Conclusion
The Conclusion is somehow very confusing. The main results should be summarized in the appropriate form and the authors must provide their inference (how the studied hypothesis fit) while answering potential questions.
L367-369. These sentences are misplaced here, therefore should be removed from the conclusion.
L378-379. The sentence is confusing, please rephrase.
Author Response
We appreciate all your comments and insights. With the reflection of your review opinions, the revised parts were indicated with red letters.
1) The Introduction seems all over the place. Is important to frame the subject at hand, however, the hypothesis tested should be better explained.
=> Introduction was revised as line 37-40. Hypotheses were also introduced as Line 91-93 (hypothesis 1) and Line 105-106 (hypothesis 2).
2) L91-92. Explain what is meant by saying "first stage dietary modification"?
=> Revised as Line 88-91.
3) How was the "main" recipe obtained/developed? There is no information on the temperature conditions during the steps of processing the puree.
=> The recipe was developed in our Lab. Preparation of puree was carried out at room temperature (22-23 oC). The procedure for puree was revised as Line 120-125.
4) L120. "was" must be corrected to "were". Please verify the verb tense throughout the manuscript.
=> We verified the verb tense and revised as Line 125.
5) L119. Explain better the selection of different mass for the xanthan gum-based samples.
=> The weight of Brand G used was based on the weight of the thickener for three uses (syrup, yogurt, pudding) determined by the manufacturer, and was also determined in accordance with the Korean packaging standard.(please refer to Line 137-139)
6) I think all the steps (in Figure 1) are described before in the text. If so, Figure 1, as it is, should be removed since is not adding new information. In this particular case, I would recommend adding a Scheme (more explicit) explaining in detail all the steps, since information like processing conditions are missing (i.e. mixing velocity; temperature etc..).
=> The Figure was deleted as your comment and the procedures was explained more detail (Line 120-126).
7) L161. The 500mL is referring to the tested volume for each sample.
=> The viscosity was measured at 500 mL for each sample. We explained in detail and revised as Line 162.
8) L170. I am not sure if the designation "height" is correct. This refers to the maximum point of hardness (N) obtained, right?
=> Height is corrected. The hardness against the quality standards of age-friendly foods was measured using the construction method KS H 4897 recommended by the Korean Industrial Standard (KS). Please refer to reference 30: Ministry of Agriculture, Food and Rural Affairs, Quality standards for aged-friendly food, Korean Industrial Standards, Sejong, Korea, KS H 4897, Revision Dec. 31, 2020. http://standard.go.kr/streamdocs/view/sd;streamdocsId=72059203773233835 (accessed on 2 April 2021).
9) L174. Is there any indication of the time after package for the commercial samples? Can this be a determinant factor regarding the comparison tests?
=> The two thickeners was stored with the companies direction and the shelf life was confirmed for 2 years before use. It is believed that this did not act as a decisive factor in the loss of antioxidant activity.
10) I am curious about the fluidity of the samples. Was the procedure mainstream as a result of the increased viscosity of some samples (like measuring 0.2 mL of sample)?
=> The amount of sample for measuring viscosity was 500 mL not 0.2 mL
11) L268. Correct the (•) symbol in the entire manuscript.
=> Revised all as “mPa·s”
12) L301-302. The decimal point form should be uniformized.
=> All decimal point forms were uniformly changed (Line 299-307).
13) L303-304. The sentence is confusing and should be revised.
=> Revised as Line 309.
14) L325. The discussion of the results could be improved. G puree results are a result of poor levels of total phenolics?
=> Antioxidant properties were reanalyzed according to statistical significance tests and revised. Especially corrected as Line 343-344 and Line 363-365.
15) The Conclusion is somehow very confusing. The main results should be summarized in the appropriate form and the authors must provide their inference (how the studied hypothesis fit) while answering potential questions. => Revised as Line 374-381.
16) L367-369. These sentences are misplaced here, therefore should be removed from the conclusion.
=> The part was removed and revised
17) L378-379. The sentence is confusing, please rephrase.
=> Revised as Line 374-381.